# Lowering *n*-6/*n*-3 Ratio as an Important Dietary Intervention to Prevent LPS-Inducible Dyslipidemia and Hepatic Abnormalities in *ob/ob* Mice

**DOI:** 10.3390/ijms23126384

**Published:** 2022-06-07

**Authors:** Seohyun Park, Jae-Joon Lee, Jisu Lee, Jennifer K. Lee, Jaemin Byun, Inyong Kim, Jung-Heun Ha

**Affiliations:** 1Department of Food Science and Nutrition, Dankook University, Cheonan 31116, Korea; sb5590@naver.com (S.P.); dlwltn970811@naver.com (J.L.); 2Department of Food and Nutrition, Chosun University, Gwangju 61452, Korea; leejj80@chosun.ac.kr; 3Food Science and Human Nutrition Department, University of Florida, Gainesville, FL 32611, USA; leejennifer@ufl.edu; 4Center for Discovery and Innovation, Hackensack Meridian Health, Nutley, NJ 07110, USA; jaemin.byun@hmh-cdi.org; 5Food and Nutrition Department, Sunchon University, Suncheon 57922, Korea; 6Research Center for Industrialization of Natural Neutralization, Dankook University, Yongin 16890, Korea

**Keywords:** obesity, polyunsaturated fatty acids, perilla oil and corn oil, inflammation, dyslipidemia, cardiac risk factor

## Abstract

Obesity is closely associated with low-grade chronic and systemic inflammation and dyslipidemia, and the consumption of omega-3 polyunsaturated fatty acids (*n*-3 PUFAs) may modulate obesity-related disorders, such as inflammation and dyslipidemia. An emerging research question is to understand the dietary intervention strategy that is more important regarding *n*-3 PUFA consumption: (1) a lower ratio of *n*-6/*n*-3 PUFAs or (2) a higher amount of *n*-3 PUFAs consumption. To understand the desirable dietary intervention method of *n*-3 PUFAs consumption, we replaced lard from the experimental diets with either perilla oil (PO) or corn oil (CO) to have identical *n*-3 amounts in the experimental diets. PO had a lower *n*-6/*n*-3 ratio, whereas CO contained higher amounts of PUFAs; it inherently contained relatively lower *n*-3 but higher *n*-6 PUFAs than PO. After the 12-week dietary intervention in *ob/ob* mice, dyslipidemia was observed in the normal chow and CO-fed *ob/ob* mice; however, PO feeding increased the high density lipoprotein-cholesterol (HDL-C) level; further, not only did the HDL-C level increase, the low density lipoprotein-cholesterol (LDL-C) and triglyceride (TG) levels also decreased significantly after lipopolysaccharide (LPS) injection. Consequently, extra TG accumulated in the liver and white adipose tissue (WAT) of normal chow- or CO-fed *ob/ob* mice after LPS injection; however, PO consumption decreased serum TG accumulation in the liver and WAT. PUFAs replacement attenuated systemic inflammation induced by LPS injection by increasing anti-inflammatory cytokines but inhibiting pro-inflammatory cytokine production in the serum and WAT. PO further decreased hepatic inflammation and fibrosis in comparison with the ND and CO. Hepatic functional biomarkers (aspartate aminotransferase (AST) and alanine transaminase (ALT) levels) were also remarkably decreased in the PO group. In LPS-challenged *ob/ob* mice, PO and CO decreased adipocyte size and adipokine secretion, with a reduction in phosphorylation of MAPKs compared to the ND group. In addition, LPS-inducible endoplasmic reticulum (ER) and oxidative stress decreased with consumption of PUFAs. Taken together, PUFAs from PO and CO play a role in regulating obesity-related disorders. Moreover, PO, which possesses a lower ratio of *n*-6/*n*-3 PUFAs, remarkably alleviated metabolic dysfunction in LPS-induced *ob/ob* mice. Therefore, an interventional trial considering the ratio of *n*-6/*n*-3 PUFAs may be desirable for modulating metabolic complications, such as inflammatory responses and ER stress in the circulation, liver, and/or WAT.

## 1. Introduction

The prevalence of global obesity has risen dramatically since 1975 [1]. According to the World Health Organization (WHO) estimates, “In 2016, more than 1.9 billion adults aged 18 years and older, were overweight. Of these over 650 million were obese [2]”. If current trends continue, it is likely that one in five adults worldwide will become obese by 2025 [3]. Obesity can occur due to reduced physical activity and increased energy intake from simple sugars and lipids. According to the WHO, body mass indices (calculated as body weight in kilograms divided by height in meters squared, BMI) >25 indicate overweight, and individuals with BMI >30 are classified as obese [4]. 

As the main storage compartment of adipose tissues (ATs), fatty acids (FAs) play a significant role in developing obesity and metabolic syndrome (MetS). MetS is considered when three or more of the following five disorders are present: (1) waist circumference >40 inches (men) or >35 inches (women), (2) blood pressure >130/85 mmHg, (3) fasting triglyceride (TG) level >150 mg/dL, (4) fasting high-density lipoprotein (HDL) cholesterol level <40 mg/dL (men) or <50 mg/dL (women), and (5) fasting blood sugar >100 mg/dL [5]. In addition to these metabolic indices, obesity-related MetS is strongly associated with endothelial dysfunction, atherogenic risk, insulin resistance, and low-grade chronic inflammation [6,7]. In the United States (US), approximately 34% of adults suffer from MetS-related obesity onset. A high prevalence of MetS is a significant factor that triggers cardiovascular diseases (CVDs), non-alcoholic fatty liver disease (NAFLD), and type 2 diabetes mellitus (T2DM) [7]. At the cellular pathophysiological level, obesity-related metabolic disturbances may also induce endoplasmic reticulum (ER) stress.

In hypertrophic obesity, enlarged adipose tissue releases excess free fatty acids (FFAs) into the blood stream, and elevated FFA levels trigger ectopic fat accumulation in multiple tissues [8]. TG accumulation as ectopic fat in the liver is caused by the esterification of FFA from the circulation and glycerol in the liver [9,10]. Hepatic ectopic fat accumulation may also be affected by increased fatty acid synthesis and delivery and/or decreased FA oxidation [11,12]. According to the prevailing “two-hit hypothesis” model, the “first hit” involves ectopic fat accumulation in the liver [13]. The “first hit” increases the vulnerability of the liver to follow up multiple pathological factors that may trigger the “second hit” and increase hepatic injury, inflammation, and fibrosis [14].

It has been widely accepted that the Mediterranean diet, an *n*-3 enriched diet, prevents or reduces the risk of CVDs [15]. Polyunsaturated fatty acid (PUFA)s are essential FAs because they cannot be synthesized in the human body and must be obtained from food sources such as sardines, herring, mackerel, smelt, tuna, anchovy, bluefish, trout, and salmon [16]. α-linolenic acid (ALA) is available from *n*-3 PUFA enriched sources such as perilla, flaxseed, and *Camelina sativa* oil [17]. Several studies have clearly demonstrated that the intake of *n*-3 PUFA may mitigate the degree of CVD risk, insulin resistance, obesity, and serum lipids in both rodents and humans [18,19]. In addition, the consumption of *n*-3 PUFAs could prevent low-grade inflammation in ATs induced in obese or diabetic mice [20,21,22]. *n*-6 PUFA-rich diets, such as those with arachidonic acid (AA) or AA-derived eicosanoids, are known to regulate a wide variety of immunopathological processes, involved in inflammation, chronic tissue remodeling, autoimmune disorders, and cancer [23]. 

Perilla oil (PO) is an edible oil extracted from *Perilla frutescens* seeds that is used for cooking in Korea and Japan. It is known to have several health-promoting effects, including neuroprotective, anti-inflammatory, and lipid-lowering effects [24]. PO is composed of several types of FAs, with 6.7–7.6% being saturated fatty acids (SFAs) and 13–20% being PUFAs [25]. Previous studies have indicated that PO consumption decreases postprandial plasma lipids in male rats [26] and decreases blood TG, total cholesterol (TC), and low density lipoprotein-cholesterol (LDL-C) levels in mice and humans [27,28]. PO is considered a source of *n*-3 PUFAs (58%), including ALA [25]. Corn oil (CO) is composed of PUFAs (59%), monounsaturated fatty acids (MUFA)s (24%), and SFAs (13%) [29], and CO has a higher proportion of PUFAs than PO. Balancing the ratio of *n*-6/*n*-3 PUFAs is strongly recommended for the normal functioning of various physiological processes. The optimal *n*-6/*n*-3 PUFA ratio should be approximately 1–2:1 for normal physiological functions in the body due to the competitiveness of *n*-6 and *n*-3 PUFAs [30,31]. ALA and linoleic acid (LA) are metabolized by competitive common enzymatic reactions; therefore, increasing the *n*-6 PUFA consumption by LA intake may inhibit eicosapentaenoic acid (EPA) and docosahexaenoic acid (DHA) synthesis from *n*-3 PUFAs. There is a growing concern regarding the *n*-6/*n*-3 ratio since the typical Western diet has a *n*-6/*n*-3 ratio of 10:1 [32].

Previously, we have reported that the partial replacement of HFD with PO or CO could alleviate high fat-induced obesity, dyslipidemia, insulin resistance, and hepatic steatosis in rats [33]. In addition, the partial replacement of dietary SFAs with PUFAs attenuates LPS-induced inflammation in the liver and metabolic complications in mice as compared with HFD consumption [34]. In this study, we raised the following research question: What is the more important factor related to *n*-3 consumption in dyslipidemia and atherogenic risks in *ob/ob* mice: (1) the ratio of *n*-6/*n*-3 PUFAs or (2) the absolute amount of *n*-3 PUFAs consumption? The *ob/ob* mice are genetically deficient in the *Ob* gene, and leptin (expressed protein of the *Ob* gene) deficiency results in extreme obesity, hyperphagia, insulin resistance, and dyslipidemia with reduced energy expenditure, and these mice are easily prone to MetS [35,36,37]. In the current study, we hypothesized that the partial replacement of SFAs with PO or CO could attenuate metabolic complications and anti-inflammatory effects after LPS injection by balancing the ratio of *n*-6/*n*-3 PUFAs or increasing the intake of *n*-3 PUFAs in genetically obese mice.

## 2. Results

### 2.1. Dietary Fatty Acid Profiles in Experimental Diets

Experimental diets were fabricated by partially replacing the fat portions with 1% PO (PO group) and 3% CO (CO group). The experimental diets of the PO and CO groups were fabricated to have identical absolute amounts of *n*-3 PUFAs, and the presence of a low amount of *n*-3 ratio in the CO group relatively to PO group increased the total PUFAs content to twice of that of the PO group. The total fat ratio in the diet was 15.75% in all the groups (Table 1). The ratio of SFAs was higher in the diet of the ND group as compared to those in the PO and CO groups (53.07% vs. 44.77% and 42.04%, respectively). The diet of the ND group had a higher content of SFAs and MUFAs than those of the PO and CO groups. Notably, levels of *n*-3 PUFAs were similar between the diets of the PO and CO groups, in line with our intention (10.44% and 10.02%, respectively; Table 2). However, the content of *n*-6 PUFAs was higher in the diet of the CO group (21.51%) than those in the ND (0.75%) and the PO (5.03%) groups. Therefore, the *n*-6/*n*-3 ratio of PUFAs was 0.49 for the ND group, 0.10 for the PO group, and 2.15 for the CO group, with the PO group having a 4.39-fold and 20-fold less *n*-6/*n*-3 ratio than the ND and CO groups, respectively (Table 2).

### 2.2. Effect of Partial Replacement of Dietary Fatty Acids with PUFAs on Body Weight (BW)

Experimental *ob/ob* mice were fed ND, ND with PO, or ND with CO for 12 weeks (*n* = 16 per group). The CO group exhibited a significant increase in BW two weeks after the dietary intervention (Figure 1A, *p* < 0.05). The final BW of the CO group significantly increased by 1.07-fold at 12 weeks as compared to the other groups, and BW change (ΔBW) increased by 1.13-fold in comparison with ND (Figure 1B). Daily food intake was significantly higher in the CO group than that in the ND group, showing a similar trend to that of ΔBW (Figure 1C, relative ratio 1.09). In contrast, the food efficiency ratio (FER) was intact after dietary intervention, although there was a significant change in ΔBW and daily food intake in the CO group (Figure 1D).

### 2.3. Alteration of Fatty Acids Composition after Dietary Intervention

Table 3 shows the changes in FA composition detected in whole blood samples of *ob/ob* mice after the 12-week dietary intervention. The proportion of SFAs in whole blood was approximately 39%. The PO group showed markedly increased *n*-3 PUFAs (ALA, EPA, docosapentaenoic acid [DPA]), and DHA by 2.73-, 13.2-, 5.92-, and 3.34-fold, respectively) compared to the CO group. Although the amounts of *n*-3 PUFAs in the experimental diets of the PO and CO groups were similar (Table 2; 10.44% and 10.02%), the amount of *n*-3 PUFAs in whole blood was higher in the PO group than that in the CO group (13.41% vs. 5.40%). The dietary content of *n*-6 PUFAs was 4.3-fold lower in the PO group than that in the CO group, whereas the blood level of *n*-6 PUFAs in the PO group was 1.65-fold lower than that in the CO group. The ratio of *n*-6/*n*-3 PUFAs in the blood was significantly higher in the ND (10.30%) and CO (5.92%) groups than that in the PO (1.45%) group. Regarding the ratios of *n*-6/*n*-3 PUFAs in the experimental diets, the PO group had 4.39-fold and 20-fold lower ratios than the ND and CO groups (Table 2), whereas in the blood, the PO group had ratios of 7.1-fold and 4.1-fold as compared to the ND and CO groups, respectively (Table 3).

### 2.4. Fatty Acids Replacement with PO or CO Maintains Intact Insulin Sensitivity in ob/ob Mice

CO replacement increased the BW (Figure 1); therefore, we assumed that the partial replacement of dietary fat with PO and CO could alter insulin sensitivity in obese mice. The oral glucose tolerance test (OGTT) showed that replacement with PO and CO did not change the fasting serum glucose level upon glucose (1 g/Kg, BW) administration (Figure 2A,B). Moreover, PO or CO replacement had no effect on insulin sensitivity in the insulin tolerance test (ITT) (Figure 2C,D). These results imply that the partial replacement of SFAs with either PO or CO had no significant effect on glucose metabolism or insulin sensitivity in our experimental settings.

### 2.5. Fatty Acids Replacement and LPS Injection did Not Alter Relative Tissue Weight

LPS were injected 24 h before sacrifice, and the relative weights of the liver and adipose tissues, including the sum of weight of ATs, epididymal AT (EAT), mesenteric AT (MAT), retroperitoneal AT (RAT), and perirenal AT (PAT) were measured. Overall, there was no significant difference in the tissue weights after dietary treatment (Figure 3).

### 2.6. PO or CO Attenuates Cardiovascular Risk Markers in LPS-Injected ob/ob Mice

To investigate whether the lipid panels were affected by PUFAs intake and/or LPS injection, we analyzed the serum levels of TG, TC, LDL-C, and HDL-C. Serum TG levels remarkably increased after LPS injection, regardless of dietary fatty acid replacement with PO or CO (Figure 4A). The TC level in the ND group was 424.38 mg/dL after LPS injection, showing a 1.67-fold increase (Figure 4B; *p* < 0.05), as compared to that in the PBS-injected group. The levels of TC in the PO and CO groups were 370 and 361.33 mg/dL, respectively, after LPS injection, and PO and CO replacements decreased TC by 12.64% and 14.86% compared to ND, respectively. LDL-C was increased by 2.01-fold in the ND group (from 148.17 to 297.25 mg/dL) and dietary FAs replacement with PO or CO significantly lowered serum LDL-C levels in comparison with the ND group (24.5% and 31.9%, respectively) after LPS-injection (Figure 4D). On the other hand, the HDL-C level was increased in the PO (118.50 mg/dL) group without LPS injection, implying that a lower ratio of *n*-6/*n*-3 PUFAs may be more important than the absolute amount of *n*-3 PUFAs consumed in the regulation of dyslipidemia in an inflamed status. Although the levels of HDL-C were decreased by LPS injection in all the groups, PO and CO replacements lead to a lesser decrease in the HDL-C level than ND after LPS injection (Figure 4C). Cardiac risk factors (CRF), calculated using TC, LDL-C, and HDL-C, had no dietary effect without LPS injection. Intriguingly, PO and CO significantly inhibited 66 and 57% of CRF induced by LPS injection as compared to ND (Figure 4E). Taken together, our results indicate that increased amounts of PUFAs by the partial replacement of SFAs with either PO or CO attenuated dyslipidemia and CVD risk in LPS-injected *ob/ob* mice. Moreover, the lower ratio of *n*-6/*n*-3 PUFAs increased HDL-C in *ob/ob* mice in the absence of LPS injection.

### 2.7. PO or CO Suppresses Systemic Inflammation in LPS-Injected ob/ob Mice

To examine the anti-inflammatory effects of PO and CO on LPS injection, pro- and anti-inflammatory cytokines were measured in the serum. LPS injection increased the levels of pro-inflammatory cytokines, such as IL-1β, TNF-α, and CXCL-1 by 11.25-fold, 20.06-fold, and 9-fold in the serum, respectively, compared to the ND group. PUFAs replacement attenuated the induction of IL-1β levels in the serum by LPS injection as compared with the ND group (19% in PO, 17% in CO, Figure 5A). PO replacement decreased LPS-induced TNF-α levels by 48.5%, but there was no TNF-α lowering effect in the CO group (Figure 5B). Notably, CO replacement increased IL-1β and TNF-α levels by 1.71- and 19.97-fold, respectively, in the absence of LPS injection. LPS injection increased the CXCL-1 levels, whereas PO and CO replacement decreased the level of CXCL-1 by 65.9% and 31.0%, respectively (Figure 5D). Our findings indicated that LPS-inducible pro-inflammatory responses were significantly decreased by the replacement of SFAs by PUFAs. On the other hand, PO and CO increased the levels of anti-inflammatory cytokine IL-10 by 4.85-fold and 5.87-fold, respectively, in comparison with the ND group after LPS injection (Figure 5C). Notably, IL-10 did not change in the ND group, irrespective of LPS injection, which may imply that SFAs in ND have a limited ability to induce anti-inflammatory cytokines, such as IL-10. In summary, LPS injection elevated pro-inflammatory cytokine levels and repressed anti-inflammatory cytokine production in *ob/ob* mice. However, the elevation of PUFAs could alleviate inflammation through both inhibition of LPS-inducible pro-inflammatory cytokines and elevation of anti-inflammatory cytokine production.

### 2.8. PO Alleviates Hepatic Lipid Accumulation and Fibrosis in LPS-Injected ob/ob Mice

Based on the aforementioned lipid panel results (Figure 4), we raised the follow-up research question of whether systemic dyslipidemia caused by LPS injection could also induce pathological stress in hepatocytes. To understand ectopic fat deposition in the liver after LPS injection, liver TG and TC levels were measured. The PO decreased TG levels by approximately 25% compared to the ND or CO groups, although there was no difference in TC levels among the dietary interventions (Figure 6A,B). Histological evaluation was performed to determine whether PO decreased hepatic pathology (Figure 6C). The NAFLD level was assessed after histological evaluation to compare the pathological levels in the liver after LPS injection. PO decreased the histological score by 27% in comparison with the ND or CO groups (Figure 6D), and the CO group had no change compared to the ND group. These results may imply that a lower ratio of *n*-6/*n*-3 PUFAs inhibits hepatic TG accumulation and inflammation; therefore, we postulate that PO may mitigate hepatic function abnormalities in LPS-injected *ob/ob* mice. These results were observed in other mouse LPS models in our previous study [33,34,38]. To study how pathological remodeling affects liver tissue function, aspartate aminotransferase (AST), alanine transaminase (ALT), and alkaline phosphatase (ALP) levels were measured. PO decreased AST and ALT activities compared to ND (Figure 7A,B). However, CO increased AST activity, but there were no significant changes in ALT activity (Figure 7A,B). ALP enzymatic activity was not significantly altered by dietary intervention in our experimental setting (Figure 7C). In summary, LPS-injected *ob/ob* mice exhibited detrimental hepatic histology, pathology, and function. PO alone prevented pathological development of the liver following LPS injection. Our findings may imply that the difference in the ratio of *n*-6/*n*-3 PUFAs is significantly important in the pathophysiology of inflammation of the liver.

### 2.9. PO Inhibits Lipid Accumulation and Inhibits Adipocyte Size, Leading to Suppression of Inflammation in EAT in Inflammatory ob/ob Mice

Adipose tissue is an active endocrine organ that regulates both energy metabolism and immune responses [39]. PO decreased TG accumulation by 15% and 22.1% in the EAT group, compared to the ND and CO groups, respectively. However, CO slightly increased TG levels and may comply with the BW gain in the CO group (Figure 1). There was no significant difference in the TC levels among the experimental groups (Figure 8A,B), as observed in the liver (Figure 6B). Increased TG levels may imply enlargement of the adipocyte size; therefore, we evaluated the EAT using histological evaluation after hematoxylin and eosin (H&E) staining (Figure 8C). PO and CO decreased adipocyte area in the ATs by 14% and 21%, respectively, in comparison with ND (Figure 8D). These results show that the replacement of dietary fat with PUFAs can inhibit lipid accumulation and decrease ATs cell size. Using the EAT tissue harvested from LPS-injected *ob/ob* mice, we performed ex vivo plantation, and then measured the cytokines to verify the anti-inflammatory effects of PUFAs in the EAT. Pro-inflammatory cytokines (IL-1β and CXCL-1) in the EAT were remarkably decreased in the PO and CO groups compared to those in the ND group (Figure 9A,B). In contrast, anti-inflammatory cytokine IL-10 levels were also decreased in the PO and CO groups compared to the ND group (Figure 9C), which was in contrast to the systemic increase in IL-10 (Figure 5C), indicating that the decrease in IL-10 secretion from the adipocytes may be related to the decreased adipocyte size (Figure 9C).

### 2.10. ER Stress and Oxidative Gene Expression in EAT from ob/ob Mice under Inflammation

Inflamed adipocytes in obese subjects may trigger other cellular stresses (i.e., ER and/or oxidative stress) via transcriptional and/or post-translational regulatory processes [40,41]. In our previous study, PO and/or CO attenuated systemic inflammation and hepatic fibrosis [34]. Therefore, we hypothesized that PO and CO could suppress inflammatory responses and inflammation-triggered ER or oxidative stress in adipocytes. Mitogen-activated protein kinases (MAPKs) signaling is regulated by an outer stimulus via phosphorylation in a post-translational manner and is involved in multiple cellular responses (i.e., inflammation, ER stress, and/or oxidative stress) [7,42,43]. PO and CO inhibited IL-6 mRNA levels by 62% and 87%, respectively, in comparison with ND (Figure 10A). TNF-α and IL-1β mRNA expression levels were lower in the CO group than those in the ND and PO groups (Figure 10B,C). PO decreased pro-inflammatory cytokines in the system and WAT, but there was no transcriptional deduction in WAT; therefore, we assume that PO may modulate WAT inflammation in a post-translational manner. PO and CO decreased the phosphorylation of extracellular signal-regulated kinase (ERK) by 0.66- and 0.41-fold, respectively, after LPS injection (Figure 11A,B). Phosphorylation of c-Jun N-terminal kinase (JNK) was also decreased by 34.0% and 59.0% by PO and CO, respectively, after LPS-injection (Figure 11A,C). These results indicated that PO and CO decreased phosphorylation of MAPKs, which was a possible signaling pathway to reduce LPS-inducible inflammatory responses in our experimental setting, as we demonstrated previously [34] (Figure 9 and Figure 10). Prolonged inflammation may trigger other pathophysiological disorders, such as ER stress; therefore, the reduction in inflammatory responses by PO and CO may decrease ER stress in WAT. However, enigmatically, the ER chaperone in the PO and CO groups increased binding immunoglobulin protein (BiP, Hsp70 chaperone) expression compared to that in the ND group after LPS injection (Figure 11D). However, C/EBP homologous protein (CHOP, nuclear ER stress regulator) was relatively lowered in PO than others (Figure 11E). Interestingly, HO-1 (oxidative-sensor protein) protein levels were also decreased in the CO group after LPS treatment (Figure 11F). PO may also possess antioxidative effects; however, structural instability may increase the oxidative demand in mitochondria [44]. These results implied that the potential underlying mechanisms mitigated by PUFAs include MAPKs, ER stress, and oxidative stress triggered by LPS injection.

## 3. Discussion

In this study, we mainly focused on the effect of partial dietary replacement of SFAs with PUFAs on metabolic disorders in a genetically obese murine model after LPS injection. To test our hypothesis, 5-week-old male *ob/ob* mice were fed ND, ND + PO, or ND + CO for 12 weeks and given either PBS or LPS injection. It is widely accepted that increasing *n*-3 PUFAs levels or reducing the total fat intake is highly recommended [45]. Consumption of high levels of PUFAs is strongly recommended, owing to their protective effects against MetS complications [46]. Our research question was to clarify which is more important, (1) the absolute intake of *n*-3 PUFAs or (2) the lower ratio of *n*-6/*n*-3 PUFAs. In this study, we determined whether the replacement of SFAs in a normal chow diet with *n*-3 PUFAs could mitigate or prevent MetS-related indices. We also aimed to further study the importance of a lower ratio of *n*-6/*n*-3 PUFAs compared to the absolute amount of *n*-3 PUFAs. The rationale for using *ob/ob* mice as an obesity-induced MetS model was that *ob/ob* mice have shown features of clinically obese patients, such as hyperphagia, obesity, hyperinsulinemia, and hyperglycemia [37], although leptin deficiency in obese humans is rare [47]. It has been reported that acute inflammatory stimuli during high-fat diet consumption further exacerbated pro-inflammatory cytokine production and morphological changes in the liver [48]. We treated *ob/ob* mice to induce inflammatory disturbances under complicated metabolic conditions.

Clinically, atherosclerosis, associated with dyslipidemia (increased TG, TG, LDL-C, and/or decreased HDL-C levels) [49], is also considered a chronic inflammatory disease [50] and is frequently observed in subjects with MetS [51]. Other pathological stimuli for atherosclerosis may include hypertension, chronic kidney disease, aging, hyperglycemia, and inflammatory responses. Previously, we reported that the LPS injection induced hyperlipidemia accompanied by increased TG, TC, HDL-C, and non-HDL-C levels and CVD expectancy in experimental rodents [33,38]. The administration of LPS can lead to rapid and transitory increases in pro-inflammatory cytokines such as TNF-α and IL-6 in mammals [52]. As a result, LPS injection stimulates acute inflammatory responses in various organs, including the kidney, brain, lung, and liver [53,54]. In our experiment, LPS injection triggered systemic dyslipidemia by increasing TG, TC, and LDL-C levels and decreasing HDL-C levels in obese mice (Figure 4). Compared to ND, PO and CO remarkably attenuated CRF in LPS-challenged obese mice. High levels of PUFAs in the blood may potentially decrease CRF by reducing dyslipidemia, but the intake of a lower *n*-6/*n*-3 ratio of PUFAs did not exert more lipid-lowering effects. Taken together, the amount of PUFA consumption is the more important regardless of the less/more ratio of n-6/*n*-3 PUFA to regulate systemic dyslipidemia in current experimental settings. Obesity is considered a chronic low-grade inflammation of the system [55]. In obese individuals, macrophages infiltrate the WAT, and adipokine secretion (i.e., TNF-α, IL-6, and IL-10) is elevated compared to the normal status. In addition, an increase in pro-inflammatory cytokines (i.e., TNF-α and IL-6) in the circulation may trigger diabetes or atherosclerosis [56,57]. Thus, inhibiting excess inflammatory responses is a therapeutic and preventive target against other inflammation-related disorders (i.e., diabetes or atherosclerosis) [58]. Serum levels of cytokines, such as IL-1β and TNF-α, increased in LPS-challenged obese mice and were lower in the PO and CO groups (Figure 5A,B). CO increased the level of serum IL-10, an anti-inflammatory cytokine, in *ob/ob* mice as a result of dietary fat replacement, compared to ND or PO. The effect of CO on increasing anti-inflammatory cytokines is thought to be because CO has more total PUFAs than the PO or ND groups. In the inflamed state, PO and CO increased IL-10 (Figure 5C). CXCL-1 is a chemokine that attracts neutrophils and is known to play a critical role in tissue inflammation [59]. The LPS-inducible increase in CXCL-1 decreased after PO and CO replacement (Figure 5D). Taken together, these results imply that PUFAs from PO and CO-replaced diets may inhibit proinflammatory reactions and increase anti-inflammatory activity. In addition, dietary lard replacement with PO attenuated LPS-induced systemic inflammation compared with CO replacement.

Dyslipidemia is a strong pathological consequence of inflammatory responses in the hepatic pathologies associated with MetS [60,61,62]. NAFLD is a type of steatosis, where >5% ectopic fat accumulates in the liver in the absence of competing etiologies such as heavy alcohol use. NAFLD may pathologically accelerate the risk of non-alcoholic steatohepatitis (NASH) through inflammation and fibrosis in the liver [63]. NASH is caused by excess fat accumulation in hepatocytes of individuals with obesity-induced MetS, associated with impaired inflammatory cytokines, adipokines, mitochondrial dysfunction, and oxidative stress. As the most severe form of NAFLD, NASH may progress to fibrosis, cirrhosis, liver failure, or hepatocellular carcinoma [64]. In the current study, both PO and CO replacement significantly lowered LPS-inducible dyslipidemia (Figure 4); therefore, we carefully examined lipid accumulation in the liver and EAT. Surprisingly, PO replacement remarkably attenuated hepatic and adipocyte TG accumulation (Figure 6A and Figure 8A).

Hepatic ectopic fat accumulation may also be affected by increased fatty acid synthesis and delivery and/or decreased FA oxidation [13,14]. According to the prevailing model of “two-hit hypothesis”, the “first hit” involves the ectopic fat accumulation in the liver [15]. The “first hit” increases the vulnerability of the liver to follow up multiple pathological factors that may trigger the “second hit” and increase hepatic injury, inflammation, and fibrosis [16]. PO only attenuated ectopic fat accumulation in the liver; therefore, we logically postulated that PO mitigates the “first hit” and subsequent stress would be mitigated, rather than other dietary interventions. As expected, PO replacement remarkably attenuated hepatic fibrosis (Figure 6C,D) and function-related indices (AST and ALT) (Figure 7A,B). However, CO replacement did not decrease hepatic TG accumulation as a “first hit” compared with ND (Figure 6A). Patients with NAFLD are recommended to have lower ratios of *n*-6/*n*-3 PUFAs than healthy people [65], since excess *n*-6 PUFAs intake may form oxidized linoleic acid metabolites (OXLAMs) in the liver, which may be a potential risk factor for fatty liver development in both youth and adults [66]. Although CO has the same amount of *n*-3 PUFA as PO, it has a higher *n*-6 level, and the ratio of *n*-6/*n*-3 PUFAs was relatively higher (Table 2). Increased intake of total PUFAs from PO or CO is beneficial for the regulation of dyslipidemia (Figure 4); however, only the lower ratio of *n*-6/*n*-3 PUFAs from the PO diet attenuated LPS-induced systemic cytokines and hepatic TG accumulation, fibrosis, and dysfunction. 

Owing to the significant suppression of systemic inflammation by PO replacement, we focused on adipocyte inflammation. Both PO and CO replacement significantly decreased the number of adipocytes in the EAT (Figure 8C,D). Therefore, we postulated that adipokine secretion would be attenuated by PO and CO replacement. As expected, PO and CO decreased adipokine secretion from the EAT in comparison with ND (Figure 9 and Figure 10), as discovered upon using the ex vivo plantation method. Notably, IL-1β and CXCL-1 protein secretion was remarkably decreased in the PO and CO groups of LPS-injected obese mice (Figure 9A,B). Subsequently, we measured cytokine mRNA expression in the EAT. As expected, IL-6 mRNA expression was significantly attenuated by PO and CO replacements (Figure 10A); however, only CO inhibited IL-1β mRNA expression (Figure 10B). Therefore, we hypothesized that PO might regulate cytokine expression in a post-translational manner. As expected, PO controlled the inflammatory response by controlling the phosphorylation of MAPKs in the EAT (Figure 11). Prolonged inflammation may lead to other cellular disorders, thereby increasing oxidative and/or ER stress. The activation of MAPK (i.e., ERK and JNK) by phosphorylation is a common denominator of inflammation (i.e., NF-κB activation) and ER stress [67,68,69]. In the present study, the phosphorylation levels of ERK and JNK were significantly downregulated by PO and CO (Figure 11A–C). In addition, PO and CO significantly decreased CHOP protein expression as ER stress markers (Figure 11A,E). These data imply that PO and CO may attenuate LPS-induced inflammation and ER stress. In our previous study, we found that lard replacement with a lower ratio of *n*-6/*n*-3 PUFAs (0.4) suppressed tunicamycin (TM)-induced ER stress in the livers of *db/db* mice fed lard replacement with other ratios of *n*-6/*n*-3 PUFAs (0.9) with phosphorylation of AMPK [33,34,38]. Taken together, the consumption of PUFAs either from PO or CO is important to decrease LPS-inducible adipocyte inflammation by measuring adipokine secretion and transcriptional and post-translational protein expression. Intriguingly, PO decreased adipokine secretion from the EAT, unlike CO.

*n*-3 PUFAs from food or supplements are essential as structural and functional nutrients to maintain normal physiology in the heart, lungs, blood vessels, and immune system [18,70,71]. Consuming *n*-3 PUFAs has been highlighted because the population with obesity-inducible metabolic complications is growing rapidly [72]. Obesity-inducible metabolic complications are directly correlated with life expectancy and quality [73]; therefore, authorities set up dietary recommendation for *n*-3 PUFAs intake (Table 4). Two major features of dietary guidance for *n*-3 PUFAs consumption are the absolute amount of *n*-3 PUFAs intake and the *n*-6/*n*-3 ratio as adequate intake (AI) (Table 4). Guidelines from Korea [74], the US [75], the European Food Safety Authority (ESFA) [76], and China [77] suggest AI levels for either *n*-3 or *n*-6 PUFAs. Korean [74] and American guidelines [75] designate the consumption amounts for linoleic acid (10–13 and 11–17 g/day for Korean and American guidelines, respectively) and α-linolenic acid (1.2–1.6 and 1.1–1.6 g/day for Korean and American guidelines, respectively). Moreover, guidelines from Korea [74] and ESFA [76] recommend AI levels for the sum of EPA and DHA (≥150 mg/day for Korean and 250 mg/day for ESFA guidelines). In addition, some guidelines suggest a recommended ratio of *n*-6/*n*-3 PUFAs: guidelines from Korea [74] and the US [75] are 4–10 and ~3, respectively. Based on the review of multiple dietary guidelines, we suggest that both the intake amount and ratio of *n*-6/*n*-3 PUFAs should be considered. Based on our experiment, the consumption of PUFAs was effective in mitigating systemic dyslipidemia and inflammatory responses. Lowering the *n*-6/*n*-3 ratio effectively reduced LPS-induced hepatic fibrosis, function, and TG accumulation in the liver and adipocytes. Therefore, by studying the consumption of PUFAs by subjects with MetS, we may better understand the pathophysiological aspects in these subjects to determine the mechanism of action of *n*-3 PUFAs consumption. There are various complicated pathological disturbances in MetS; therefore, in future studies, we should scrutinize the correlations between the consumption of *n*-3 PUFAs and other metabolic stresses (i.e., ER stress, oxidative stress, etc.).

## 4. Materials and Methods

### 4.1. Animal Experiments and Diets

All animal studies were approved by the Institutional Animal Care and Use Committee of Dankook University (IACUC, No. DKU-19-036) on 12 November, 2019. A total of 48 5-week-old male *ob/ob* mice were obtained from DooYeol Biotech (Seoul, Korea). PO and CO (CJ Cheil Jedang Co., Seoul, Korea) were obtained from a local market in Korea. Experimental mice were maintained under controlled temperature of 22 ± 1 °C and humidity of 55 ± 5% with a 12 h light/12 h dark cycle. Animals were randomly assigned to one of three dietary groups (*n* = 16 per group): (1) ND group, (2) ND partially replaced with 1% (*w*/*w*) PO, and (3) ND partially replaced with 3% (*w*/*w*) CO. The percentage of calories from fat was 15.75% in all the groups. The diet compositions are listed in Table 1. All animals had ad libitum access to their diet and purified water. BW and food consumption were measured once per week. The ΔBW was calculated by subtracting the initial BW from the final BW. Daily food intake was calculated using the following formula: (1 week of food intake/7 days). The FER was determined using the following formula: (total food intake/ΔBW). Before sacrifice, each group was further divided into two subgroups to receive the following: (1) intraperitoneal injection of LPS (*Escherichia coli*, O55:B5, Sigma, St. Louis, MO, USA, 1 mg/kg body weight) or (2) intraperitoneal injection of phosphate-buffered saline (PBS) as a control. After 24 h of PBS or LPS injection, the mice were sacrificed by thoracotomy after CO_2_ narcosis. Blood was collected by cardiac puncture and centrifuged at 1000× *g* at 4 °C for 15 min to obtain serum. Liver and adipose tissues, including EAT, MAT, RAT, and PAT, were collected and weighed. These tissues and serum samples were stored at −80°C until further analyses.

### 4.2. Analysis of Fatty Acids of Experimental Diets

The methods used for the analysis of FA have been previously described [33,34,38]. The FA composition of the experimental diets was determined after methyl esterification of boron trifluoride (BF_3_)–methanol [38]. Briefly, 0.1 g of each sample was placed in a test tube and mixed with 0.5 mL of heptadecanoic acid (C17:0; 1 mg/mL hexane). After adding NaOH–methanol (2 mL 0.5 N NaOH–methanol), the mixture was heated at 110 °C for 10 min. Once cooled to room temperature, 4 mL of BF_3_–methanol was added, and the mixture was reheated at 110 °C for 1 h. Subsequently, 2 mL of hexane was added and mixed well using a vortex mixer for 1 min. The hexane layer in the mixture was used for lipid analysis using gas chromatography (GC) (Agilent Technologies 6890N, Agilent Technologies, CA, USA). FAs were analyzed based on peak retention times. These were identified by referring to the peak of Supelco 37-component FAME mix (Sigma-Aldrich Co., St. Louis, MO, USA) as the standard solution. Data are expressed as percentile numbers.

### 4.3. Oral Glucose Tolerance Test (OGTT) and Insulin Tolerance Test (ITT)

After 11 weeks of dietary intervention, OGTT was performed in mice after fasting for 12 h. Mice were orally administered glucose solution (Sigma, St. Louis, MO, USA, 1 g/kg body weight). Blood was collected at 0, 15, 30, 60, and 120 min after glucose loading. Blood glucose levels were analyzed using Accu-Chek Instant Test Strips and an Accu-Chek Instant Blood Glucose Meter (Accu-Chek, Seoul, Korea). The ITT was conducted at the end of the study. After overnight fasting, the mice were intraperitoneally injected with insulin (Human insulin, Sigma, St. Louis, MO, USA) at a dose of 1 U/kg BW. Blood glucose levels were measured at 15, 30, 60, and 120 min after insulin injection. The OGTT and ITT results were evaluated by calculating the area under the curve (AUC).

### 4.4. Determination of Serum Metabolic Parameters

Hepatic biological function was determined by measuring serum ALT, AST, ALP using commercial kits (ALT Assay Kit, AST Assay Kit, ALP Assay Kit, Embiel, Gunpo, Korea), and TC, HDL-C, and TG levels were measured with commercial kits (TC Assay Kit, HDL-C Assay Kit, TG Assay Kit, Embiel, Gunpo, Korea) as previously described [78]. LDL-C levels were calculated by subtracting HDL-C and TG values from the TC levels. The cardiovascular risk factor was calculated using the following formula: (CRF = TC/HDL-C).

### 4.5. Analysis of the Fatty Acids Composition of Whole Blood

A drop of whole blood from each mouse was collected onto a blood spot card (OmegaQuant Analytics, Sioux Falls, SD, USA) pretreated with a multicomponent antioxidant cocktail. Compositions of whole blood FAs were analyzed using a GC, as previously described [79]. Data are expressed as a percentage of the total analyzed FAs. The methods have been previously described in detail [33,34,38].

### 4.6. Measurement of Lipid Contents in the Liver and Adipose Tissue

Lipids were extracted from the liver and EAT as previously described by Bligh and Dyer [80], with slight modifications. Briefly, tissues (approximate 0.1 g) were added to a chloroform:methanol (1:2, *v*/*v*) solution and centrifuged at 805× *g* for 15 min. Subsequently, the lower phase was separated and 1 mL of n-hexane:isopropanol (3:2, *v*/*v*) was added to dissolve lipids. TG and TC levels were measured as described above and normalized by pre-measured tissue weights. The detailed methods have been described in previous studies [33,34,38].

### 4.7. Measurement of Adipokine Secretion Ex Vivo

The EAT tissues (~0.1 g) were placed in a 12-well plate (SPL, Pocheon, Korea). In each well, 2 mL of Dulbecco’s Modified Eagle Medium (DMEM; Gibco, New York, NY, USA) supplemented with 10% FBS (Gibco, NY, USA), 1% gentamicin (Gibco, NY, USA), and 1% penicillin (Gibco) was added. The tissues were incubated at 37 °C in a 5% CO_2_ incubator for 24 h. The supernatant was harvested from the culture plate and centrifuged at 500× *g* for 5 min at 4 °C. Cytokines from the EAT ex vivo culture were measured using a MILLIPLEX map Luminex assay (Millipore, Danvers, MA, USA) according to the manufacturer’s protocol.

### 4.8. Histological Evaluation of the Liver and Epididymal Adipose Tissue

The liver and ATs were fixed with 10% formalin. Tissues were embedded in paraffin blocks and sectioned at a thickness of 3–4 µm using a microtome (Leica CM1800; Wetzler, Germany). The sections were stained with H&E and representative images of the samples were taken using an optical microscope (ZEISS Axio Imager 2, Carl Zeiss, Oberkochen, Germany). Histological changes in the liver were randomly selected 20X fields for each experimental mice (*n* = 3). Histological scores were measured using Kleiner’s histological scoring system [81], which quantifies the degree of inflammatory cell infiltration, steatosis, and balloon cell infiltration. Clinical scoring included steatosis (0–3 scores), lobular inflammation (0–2 scores), and ballooning (0–2 scores) [81]. Based on the sum of scores, we determined the following: (1) NASH, sum of scores ≥ 5; (2) NASH borderline, sum of scores = 3–4; and (3) not NASH, sum of scores = 0–2. Adipocyte size (mm^2^) was determined in three randomly selected 20X fields for each experimental mice. The adipocyte size was measured using ImageJ software (NIH, Bethesda, MD, USA). The overall methods were used in previous publications [34,82].

### 4.9. Quantitative Real-Time Polymerase Chain Reaction (qRT-PCR)

qRT-PCR was performed according to previously described methods, and relative mRNA expression was presented [33,34,38]. Total RNAs were isolated from the liver and epididymal adipose tissues using NucleoZoL reagent (Macherey-Nagel, GmbH & Co. KG; Düren, Germany). Complementary DNA (cDNA) was synthesized with 1 µg of messenger ribonucleic acid (mRNA) using an iScript™ cDNA Synthesis Kit (Bio-Rad Laboratories, CA, USA). To perform quantitative polymerase chain reaction (qPCR), cDNA was mixed with iQ™ SYBR^®^ Green Supermix (Bio-Rad Laboratories). qRT-PCR was performed using the CFX96 Real-Time PCR Detection System (Bio-Rad Laboratories). The relative expression of each gene was analyzed using the 2^−ΔΔCT^ method. Mouse *Gapdh* was used as a reference gene. Primers specific for each gene used in the qRT-PCR are listed in Table 5.

### 4.10. Western Blot Analysis

The expression of the proteins of interest was determined by peer-reviewed methods [33,34,38]. Protein lysates were harvested using a homogenizer (Branson 450 digital Sonifier, Branson, Danbury, CT, USA) from frozen liver tissues in ice-cold radioimmune precipitation assay (RIPA) lysis buffer (ATTO, Tokyo, Japan) supplemented with protease and phosphatase inhibitors (Thermo Fisher Scientific, Waltham, MA, USA). Equal amounts (30 μg) of protein were separated using 10% sodium dodecyl sulfate-polyacrylamide gel electrophoresis (SDS-PAGE) and transferred to polyvinylidene difluoride (PVDF) membranes (Bio-Rad Laboratories, Hercules, CA, USA). The membranes were blocked with 5% skim milk (BD Difco, Franklin Lakes, NJ, USA). The primary antibodies were diluted in Tris-buffer and incubated overnight at 4 °C according to each concentration. After washing, the membranes were incubated with secondary antibodies conjugated to HRP at room temperature for 2 h. Proteins on the membranes were detected with an enhanced chemiluminescence solution (Thermo Fisher Scientific, Waltham, MA, USA). The protein bands were captured using a Davinch Chemi Fluoro Imager (Davinch-K, Seoul, Korea). Visualized protein bands were then quantitatively analyzed using ImageJ software (v.1.8 National Institutes of Health, Bethesda, MD, USA), with GAPDH as an internal control. Details of the antibodies are shown in Table 6. 

### 4.11. Statistical Analysis

The results are expressed as means and standard deviations (SD) or Box-and-Whisker plots. SPSS 26.0 (Statistical Package for Social Science, IBM Corp., Armonk, NY, USA) was used to test the significance of the dietary intervention using one-way analysis of variance (ANOVA). Box-and-Whisker plots depicted with the minimum, the lower (25th percentile), the median (50th percentile), the upper (75th percentile), and the maximum ranked sample and the average of the assigned group is indicated by a “+” sign. Two-way ANOVA followed by Tukey’s post hoc test was performed using XLAST 2012 (Addinsoft Inc., Paris, France) to test the relationship between fat replacement and LPS in each experimental setting. Statistical significance was set at *p* < 0.05. Table 7 presents a summary of the statistical analyses.

## 5. Conclusions

This study was designed to understand whether ratios of *n*-6/*n*-3 or the amount of *n*-3 are important by replacement of dietary lard with PO and CO. The partial replacement of dietary FAs with PUFAs attenuated dyslipidemia and inflammatory responses in LPS-induced *ob/ob* mice. Notably, PO decreased TG accumulation in the liver and adipocytes, hepatic fibrosis, and hepatic dysfunction in LPS-induced *ob/ob* mice. Therefore, a lower ratio of *n*-6/*n*-3 PUFAs may be a desirable dietary intervention for the consumption of *n*-3 PUFAs. 

## Figures and Tables

**Figure 1 ijms-23-06384-f001:**
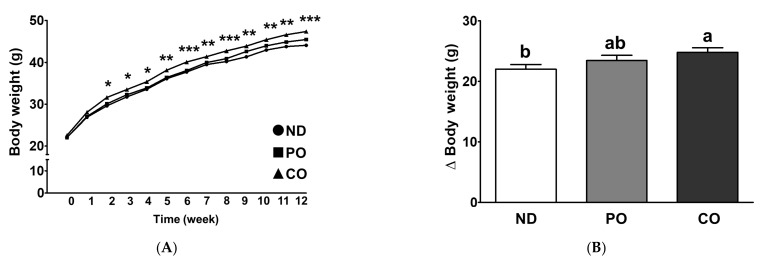
Partial replacement of dietary fat with PO or CO on body weight change, food intake, energy intake and food efficiency ratio in *ob/ob* mice. Mice were fed either an ND or an ND partially replaced with PO or CO for 12 weeks (*n* = 16 per group). (**A**) Body weight changes; (**B**) body weight gain (final BW—initial BW); (**C**) daily food intake; (**D**) food efficiency ratio. Values are presented as the mean ± standard deviation. Data were analyzed using one-way ANOVA followed by Tukey’s post hoc test. *, ** and *** denotes a significant main effect of diet at *p* < 0.05, *p* < 0.01 and *p* < 0.001, respectively. Labeled means without a common letter differ (*p* < 0.05). ND, normal diet; PO, perilla oil replacement; CO, corn oil replacement; BW, body weight.

**Figure 2 ijms-23-06384-f002:**
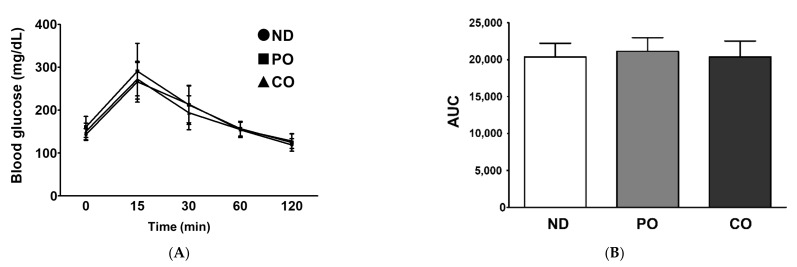
Partial replacement of dietary fat with PO or CO on the fasting glucose and insulin levels in *ob*/*ob* mice. Mice were fed either an ND or an ND partially replaced with PO or CO for 12 weeks (*n* = 16 per group). Mice were injected glucose by oral gavage and blood glucose levels were measured at 0, 15, 30, 60 and 120 min after glucose injection; oral glucose tolerance test (OGTT) (1 g/kg body weight). Insulin tolerance test (ITT) was performed after feeding the experimental diet and blood glucose levels were measured at 15, 30, 60, 120 min after insulin administration (1 U/kg body weight). (**A**) OGTT at 11 weeks; (**B**) area under the curve (AUC) of OGTT; (**C**) ITT at 11 weeks; (**D**) AUC of ITT. Values are presented as the mean ± standard deviation. Data were analyzed using one-way ANOVA followed by Tukey’s post hoc test.

**Figure 3 ijms-23-06384-f003:**
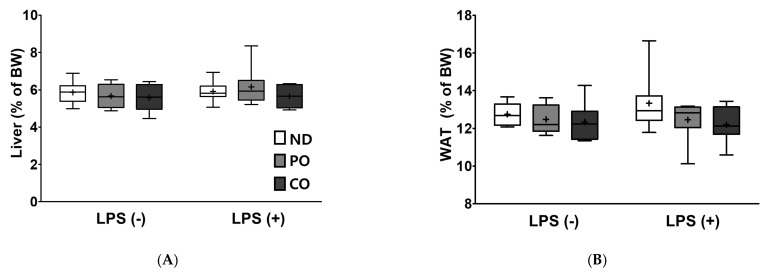
Partial replacement of dietary fat with PO or CO and LPS challenge on the liver and adipose tissue weights in *ob/ob* mice. Mice were fed either an ND or an ND partially replaced with PO or CO for 12 weeks and then treated with PBS or LPS (1 mg/kg) for 24 h (*n* = 8 per group). (**A**) Liver weight; (**B**) white adipose tissue (WAT) weight; (**C**) epididymal adipose tissue (EAT) weight; (**D**) mesenteric adipose tissue (MAT) weight; (**E**) retroperitoneal adipose tissue (RAT) weight; (**F**) perirenal adipose tissue (PAT) weight. Values are presented as box and whisker plots representing 8 mice per group. Data were analyzed using two-way ANOVA followed by Tukey’s post hoc test (LPS × Diet interaction). LPS, lipopolysaccharide; PBS, phosphate-buffered saline.

**Figure 4 ijms-23-06384-f004:**
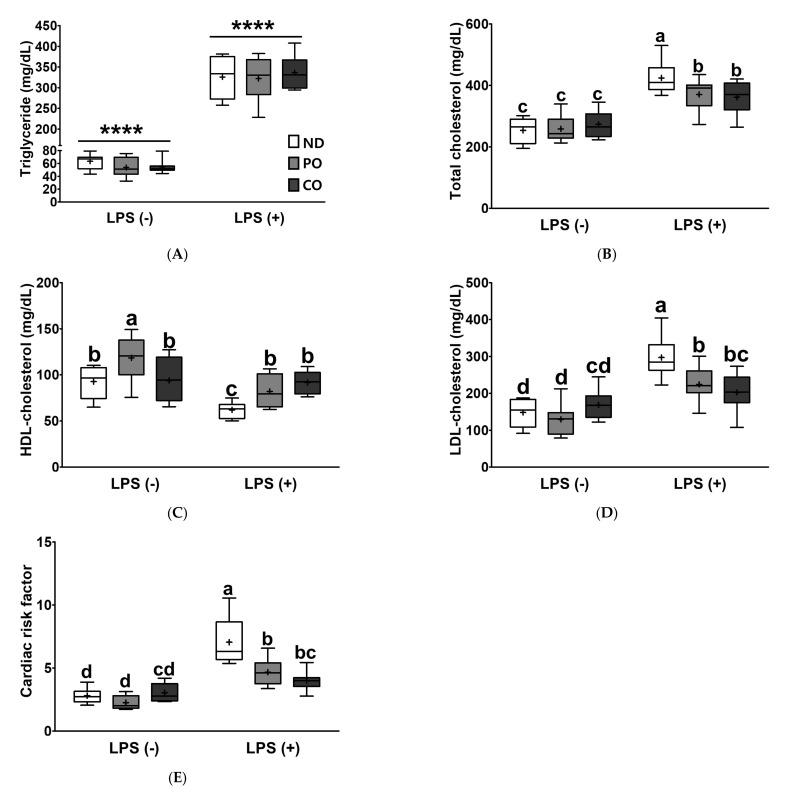
Partial replacement of dietary fat with PO or CO and LPS challenge on serum lipid profiles in *ob/ob* mice. Mice were fed either an ND or an ND partially replaced with PO or CO for 12 weeks and then treated with PBS or LPS (1 mg/kg) for 24 h (*n* = 8 per group). (**A**) Triglyceride levels; (**B**) total cholesterol levels; (**C**) high density lipoprotein (HDL)-cholesterol levels; (**D**) low density lipoprotein (LDL)-cholesterol levels; (**E**) cardiac risk factor (CRF). Values are presented as box and whisker plots representing 8 mice per group. Data were analyzed using two-way ANOVA followed by Tukey’s post hoc test (LPS × Diet interaction). Asterisk indicates a significant main effect for LPS (*****p* < 0.0001). Labeled means without a common letter differ (*p* < 0.05).

**Figure 5 ijms-23-06384-f005:**
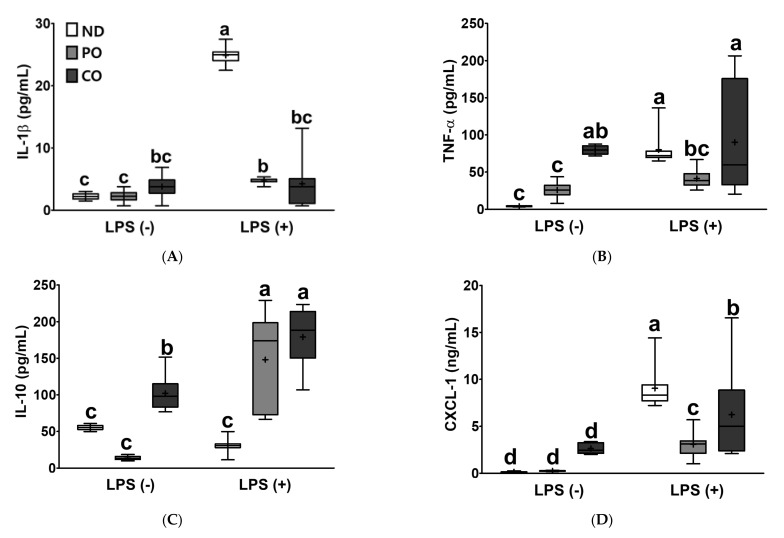
Partial replacement of dietary fat with PO or CO and LPS challenge on the pro/anti-inflammatory cytokines, chemokine in serum. Mice were fed either an ND or an ND partially replaced with PO or CO for 12 weeks and then treated with PBS or LPS (1 mg/kg) for 24 h (*n* = 8 per group). (**A**) Interleukin (IL)-1β levels; (**B**) tumor necrosis factor (TNF)-α levels; (**C**) IL-10 levels; (**D**) C-X-C Motif Chemokine Ligand 1 (CXCL-1) levels. Values are presented as box and whisker plots representing 8 mice per group. Data were analyzed using two-way ANOVA followed by Tukey’s post hoc test (LPS × Diet interaction). Labeled means without a common letter differ (*p* < 0.05).

**Figure 6 ijms-23-06384-f006:**
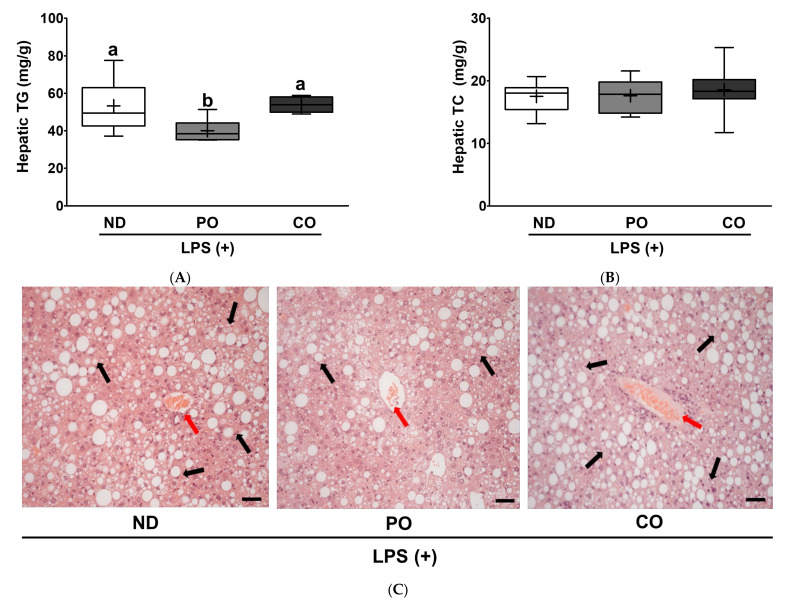
Partial replacement of dietary fat with PO or CO and LPS challenge on lipid contents and histology analysis in liver. Mice were fed either an ND or an ND partially replaced with PO or CO for 12 weeks and then treated with LPS (1 mg/kg) for 24 h (*n* = 8 per group). (**A**) Hepatic triglyceride (TG) levels; (**B**) hepatic total cholesterol (TC) levels; (**C**) H&E staining of liver tissue (bar = 50 μm, 20 × magnification); (**D**) histological scores of liver tissue. Values are presented as box and whisker plots representing 8 mice per group. Data were analyzed using one-way ANOVA followed by Tukey’s post hoc test. Black arrow represents lipid droplet and red arrow represents lobular inflammation. Labeled means without a common letter differ (*p* < 0.05).

**Figure 7 ijms-23-06384-f007:**
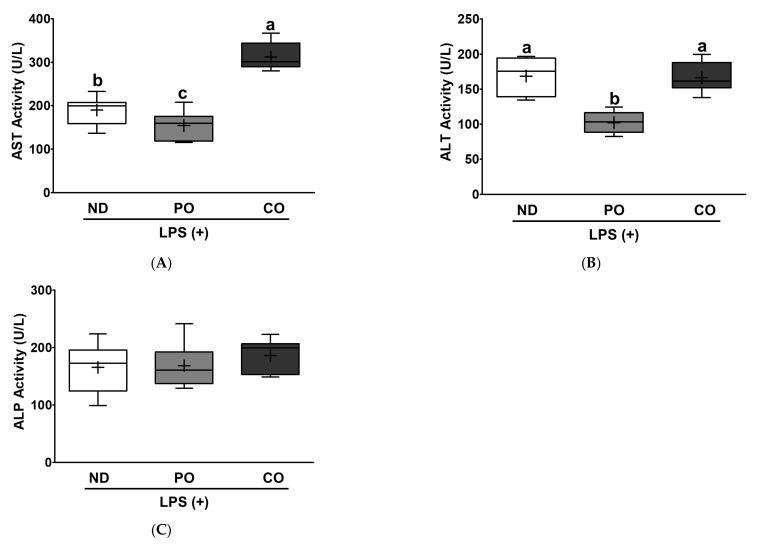
Partial replacement of dietary fat with PO or CO and LPS challenge on hepatic function parameters in serum. Mice were fed either an ND or an ND partially replaced with PO or CO for 12 weeks and then treated with LPS (1 mg/kg) for 24 h (*n* = 8 per group). (**A**) Aspartate aminotransferase (AST) activity; (**B**) alanine aminotransferase (ALT) activity; (**C**) alkaline phosphatase (ALP) activity. Values are presented as box and whisker plots representing 8 mice per group. Data were analyzed using one-way ANOVA followed by Tukey’s post hoc test. Labeled means without a common letter differ (*p* < 0.05).

**Figure 8 ijms-23-06384-f008:**
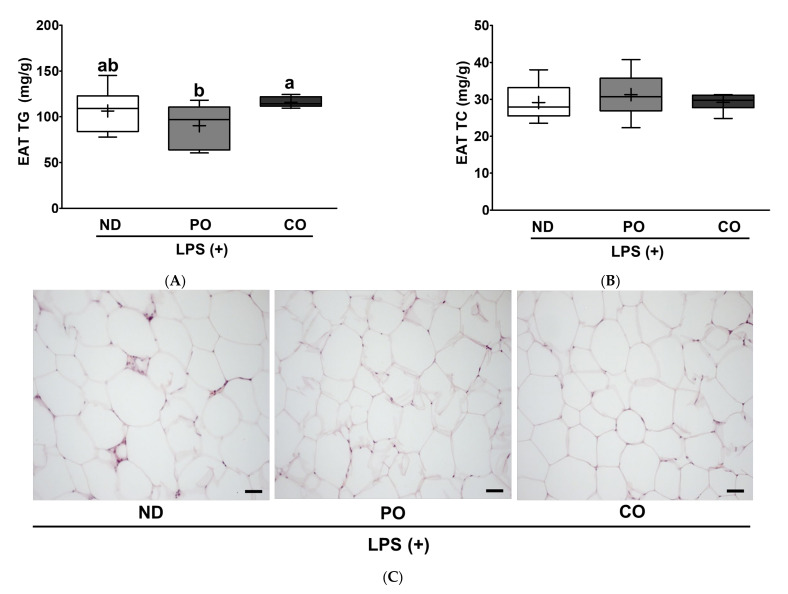
Partial replacement of dietary fat with PO or CO and LPS challenge on lipid contents and histology analysis in EAT. Mice were fed either an ND or an ND partially replaced with PO or CO for 12 weeks and then treated with LPS (1 mg/kg) for 24 h (*n* = 8 per group). (**A**) Triglyceride (TG) levels in EAT; (**B**) total cholesterol (TC) levels in EAT. (**C**) H&E staining of adipose tissue (bar = 50 μm, 20× magnification); (**D**) adipocyte area (mm^2^). Values are presented as box and whisker plots representing 8 mice per group. Data were analyzed using one-way ANOVA followed by Tukey’s post hoc test. Labeled means without a common letter differ (*p* < 0.05).

**Figure 9 ijms-23-06384-f009:**
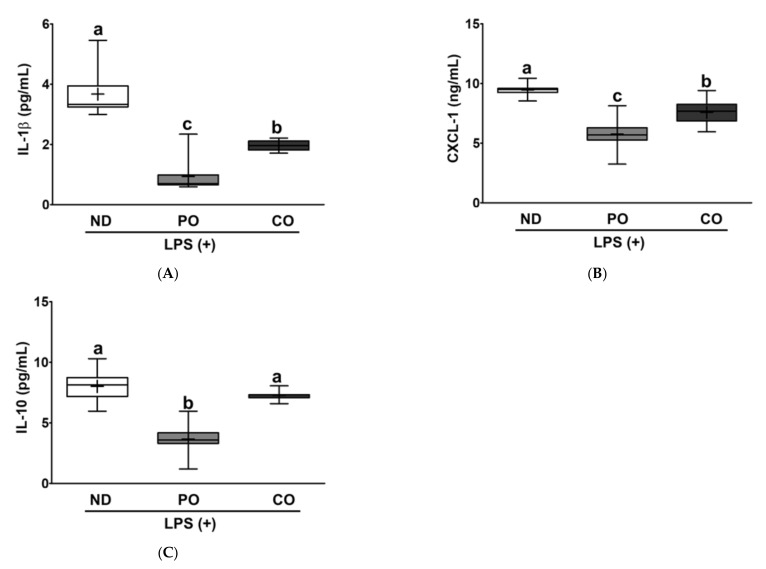
Partial replacement of dietary fat with PO or CO and LPS challenge on pro/anti-inflammatory cytokines, chemokine in EAT. Mice were fed either an ND or an ND partially replaced with PO or CO for 12 weeks and then treated with LPS (1 mg/kg) for 24 h (*n* = 8 per group). (**A**) Interleukin (IL)-1β levels; (**B**) C-X-C Motif Chemokine Ligand 1 (CXCL-1) levels; (**C**) IL-10 levels. Values are presented as box and whisker plots representing 8 mice per group. Data were analyzed using one-way ANOVA followed by Tukey’s post hoc test. Labeled means without a common letter differ (*p* < 0.05).

**Figure 10 ijms-23-06384-f010:**
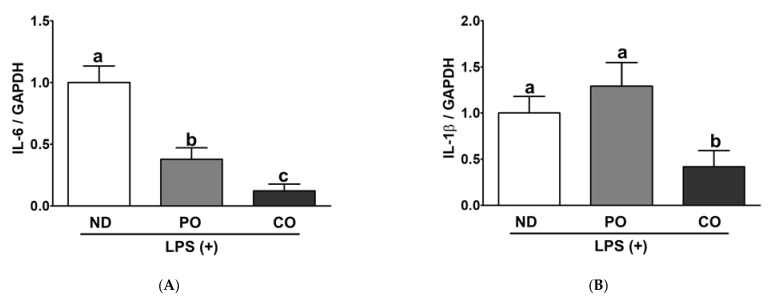
Partial replacement of dietary fat with PO or CO and LPS challenge on mRNA expression related to inflammation in EAT. Mice were fed either an ND or an ND partially replaced with PO or CO for 12 weeks and then treated with LPS (1 mg/kg) for 24 h (*n* = 8 per group). (**A**) Interleukin (IL)-6 levels; (**B**) IL-1β levels; (**C**) tumor necrosis factor (TNF)-α levels. The expression of each protein was normalized to a value for glyceraldehyde 3 phosphate dehydrogenase (GAPDH). Relative expression of each gene was quantified by using the 2^−ΔΔCT^ method. Values are presented as the mean ± standard deviation. Data were analyzed using one-way ANOVA followed by Tukey’s post hoc test. Labeled means without a common letter differ (*p* < 0.05).

**Figure 11 ijms-23-06384-f011:**
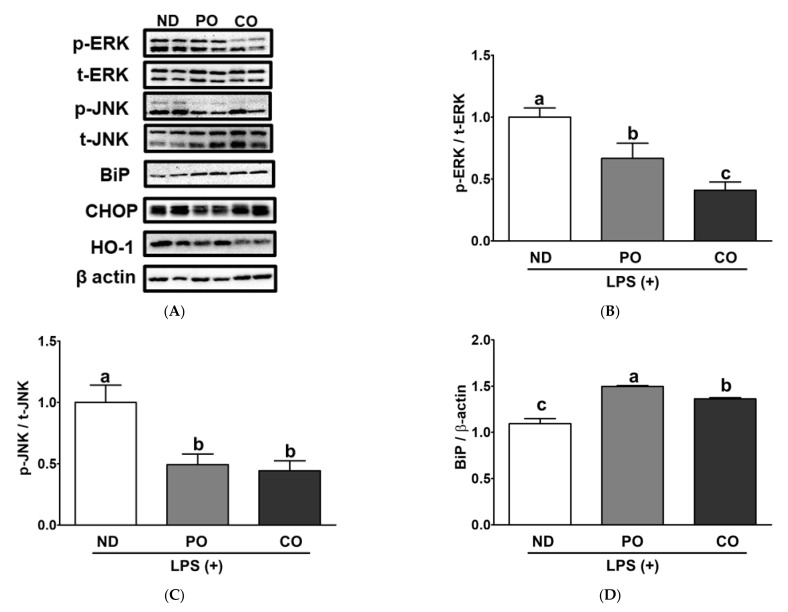
Partial replacement of dietary fat with PO or CO and LPS challenge on the protein expression of inflammatory responses in EAT. Mice were fed either an ND or an ND partially replaced with PO or CO for 12 weeks and then treated with LPS (1 mg/kg) for 24 h (*n* = 8 per group). (**A**) Representative Western blot images; (**B**) phospho-extracellular signal-regulated kinase (p-ERK) levels; (**C**) phospho-c-Jun N-terminal kinase (p-JNK) levels; (**D**) binding immunoglobulin protein (BiP) levels; (**E**) C/EBP homologous protein (CHOP) levels; (**F**) heme oxygenase 1 (HO-1) levels. The expression of each protein was normalized to a value for β-actin, the internal control of protein content. Values are presented as the mean ± standard deviation. Data were analyzed using one-way ANOVA followed by Tukey’s post hoc test. Labeled means without a common letter differ (*p* < 0.05).

**Table 1 ijms-23-06384-t001:** Composition of experimental diets.

	ND (g)	PO (g)	CO (g)
Casein	200	200	200
L-cysteine	3	3	3
Sucrose	100	100	100
Cornstarch	397.5	397.5	397.5
Dextrose	132	132	132
*tert*-Butylhydroquinone	0.014	0.014	0.014
Cellulose	50	50	50
Mineral mix	35	35	35
Vitamin mix	10	10	10
Choline bitartrate	2.5	2.5	2.5
Western blend	50	45	30
Lard	20	15	10
PO	-	10	-
CO	-	-	30
Total (g)	1000.014	1000.014	1000.014
Total energy (kcal/g)	4000	4000	4000
Energy from fat (kcal%)	15.75	15.75	15.75

ND, normal diet; PO, perilla oil replacement; CO, corn oil replacement; - not applicable.

**Table 2 ijms-23-06384-t002:** Fatty acids profile of experiment diets.

	ND (%)	PO (%)	CO (%)
Palmitic acid (C16:0)	37.61 ± 0.93 ^a^	33.86 ± 2.60 ^b^	33.26 ± 2.11 ^b^
Stearic acid (C18:0)	15.46 ± 3.58 ^a^	10.91 ± 0.75 ^b^	8.78 ± 0.61 ^b^
Elaidic acid (C18:1*n*-9t)	4.23 ± 0.94 ^a^	4.18 ± 0.12 ^a^	1.76 ± 0.10 ^b^
Oleic acid (C18:1*n*-9c)	40.30 ± 1.39 ^a^	36.61 ± 1.00 ^b^	24.66 ± 1.48 ^c^
Linoleic acid (C18:2*n*-6c)	0.75 ± 0.44 ^c^	5.03 ± 0.76 ^b^	21.51 ± 0.14 ^a^
α-linolenic acid (C18:3*n*-3)	0.82 ± 0.20 ^b^	10.44 ± 0.89 ^a^	10.02 ± 0.72 ^a^
SFA (%)	53.07	44.77	42.04
MUFA (%)	44.78	40.79	26.42
PUFA (%)	2.02	15.47	31.53
*n*-3 (%)	1.52	10.44	10.02
*n*-6 (%)	0.75	5.03	21.51
Total fatty acid (%)	100.00	100.00	100.00

ND, normal diet; PO, perilla oil replacement; CO, corn oil replacement; SFA, saturated fatty acids; MUFA, monounsaturated fatty acids; PUFA, polyunsaturated fatty acids; *n*-3, omega-3 fatty acids; *n*-6, omega-6 fatty acids. Labeled means without a common letter differ (*p* < 0.05). Unit is expressed as percentage (%) from the total fatty acids.

**Table 3 ijms-23-06384-t003:** Fatty acids composition of the whole blood of mice.

	ND (%)	PO (%)	CO (%)
α-linolenic acid	0.11 ± 0.07 ^b^	0.3 ± 0.05 ^a^	0.04 ± 0.02 ^c^
Eicosapentaenoic acid (EPA)	0.27 ± 0.12 ^b^	3.57 ± 0.36 ^a^	0.35 ± 0.09 ^b^
Docosapentaenoic acid (DPA)	0.24 ± 0.10 ^b^	1.42 ± 0.15 ^a^	0.26 ± 0.05 ^b^
Docosahexaenoic acid (DHA)	2.43 ± 1.97 ^c^	8.12 ± 0.41 ^a^	4.76 ± 0.42 ^b^
Linoleic acid	6.09 ± 0.91 ^c^	7.06 ± 0.23 ^b^	8.07 ± 0.95 ^a^
γ-linolenic acid	0.04 ± 0.01 ^b^	0.04 ± 0.02 ^b^	0.06 ± 0.01 ^a^
Eicosadienoic acid	0.22 ± 0.21 ^a^	0.14 ± 0.04 ^a^	0.15 ± 0.03 ^a^
Dihomo-γ-linolenic acid	1.36 ± 0.56 ^a^	1.42 ± 0.07 ^a^	1.56 ± 0.17 ^a^
Arachidonic acid	12.41 ± 4.81 ^b^	10.20 ± 0.27 ^b^	20.20 ± 1.71 ^a^
Docosatetraenoic acid	0.55 ± 0.19 ^b^	0.33 ± 0.02 ^c^	1.06 ± 0.10 ^a^
Docosapentaenoic acid	0.42 ± 0.17 ^b^	0.08 ± 0.05 ^c^	0.67 ± 0.10 ^a^
Oleic acid	30.15 ± 7.11 ^a^	23.55 ± 0.90 ^b^	19.45 ± 2.24 ^b^
Eicosenoic acid	0.32 ± 0.03 ^a^	0.24 ± 0.02 ^b^	0.24 ± 0.01 ^b^
Nervonic acid	0.06 ± 0.03 ^a^	0.08 ± 0.02 ^a^	0.05 ± 0.01 ^a^
Palmitoleic acid	7.44 ± 3.98 ^a^	3.88 ± 0.70 ^b^	3.29 ± 0.92 ^b^
Myristic acid	1.36 ± 0.83 ^a^	0.70 ± 0.20 ^b^	0.66 ± 0.38 ^b^
Palmitic acid	25.73 ± 1.29 ^c^	28.50 ± 0.85 ^a^	27.10 ± 1.03 ^b^
Stearic acid	11.88 ± 1.96 ^a^	9.67 ± 0.10 ^b^	11.15 ± 1.22 ^a^
Lignoceric acid	0.20 ± 0.07 ^a^	0.17 ± 0.05 ^a^	0.20 ± 0.03 ^a^
*n*-3	2.05 ± 2.10 ^b^	13.41 ± 0.60 ^a^	5.40 ± 0.47 ^b^
*n*-6	21.09 ± 6.71 ^b^	19.27 ± 0.35 ^b^	31.77 ± 1.40 ^a^
*n*-9	30.53 ± 7.12 ^a^	23.86 ± 0.89 ^b^	19.76 ± 2.25 ^b^
SFA	39.16 ± 2.40 ^b^	39.03 ± 0.77 ^a^	39.11 ± 1.67 ^a^
MUFA	37.97 ± 11.10 ^a^	27.74 ± 0.42 ^b^	23.03 ± 3.14 ^b^
*n-6/n-3*	10.30 ± 0.48 ^a^	1.45 ± 0.07 ^c^	5.92 ± 0.33 ^b^
AA/EPA	48.13 ± 0.48 ^b^	2.9 ± 0.42 ^c^	60.98 ± 1.67 ^a^

ND, normal diet; PO, perilla oil replacement; CO, corn oil replacement; EPA, eicosapentaenoic acid, DPA, docosapentaenoic acid; DHA, docosahexaenoic acid; SFA, saturated fatty acids; *n*-3, omega-3 fatty acids; *n*-6, omega-6 fatty acids; MUFA, monounsaturated fatty acids; PUFA, polyunsaturated fatty acids; AA, arachidonic acid. Labeled means without a common letter differ (*p* < 0.05). Unit is expressed as percentage (%) from the total fatty acids.

**Table 4 ijms-23-06384-t004:** Recommendations for the dietary *n*-3 consumption.

Authority	Dietary Fat	Recommendation
	fat	25–35%
American Heart Association (AHA)	*n*-3 fatty acids	- Consumption of fish or shellfish 1–2 times per week (250 mg/day calculated with EPA + DHA level) - Daily intake of EPA + DHA for healthy people (500 mg/day) and for CVD patient (800–1000 mg/day)
U.S.	fat linoleic acid alpha-linolenic acid (ALA) *n*-3 fatty acids	20–35% 11–17 g/day 1.1–1.6 g/day 0.6–1.2% of total energy intake
Europe	Fat	Less than 30%
		SFA less than 10%
		Trans fat less than 1%
FAO	PUFAs	6–10 % of total energy intake
	*n*-3 fatty acids	1–2%
	*n*-6 fatty acids	5–8%
Institute of Medicine (IOM)	*n*-3 fatty acids	~1.6 g/day
	*n*-6 fatty acids	~17 g/day

AHA, American Heart Association; *n*-3, omega-3 fatty acids; EPA, eicosapentaenoic acid, DHA, docosahexaenoic acid; CVD, cardiovascular disease; ALA, alpha-linolenic acid; SFA, saturated fatty acids; FAO, Food and Agriculture Organization of the United Nations; PUFA, polyunsaturated fatty acids; *n*-6, omega-6 fatty acids; IOM, Institute of Medicine.

**Table 5 ijms-23-06384-t005:** qRT-PCR primer sequences (5′ to 3′).

Transcript	Forward	Reverse
*Pparγ*	GGC GAT CTT GAC AGG AAA GAC	CCC TTG AAA AAT TCG GAT GG
*Cebpα*	GGT TTT GCT CTG ATT CTT GCC	CGA AAA AAC CCA AAC ATC CC
*aP2*	AGC ATC ATA ACC CTA GAT GGC G	CAT AAC ACA TTC CAC CAC CAG C
*Pgc1α*	CCC TGC CAT TGT TAA GAC C	TGC TGC TGT TCC TGT TTT C
*Il-6*	CTG CAA GAG ACT TCC ATC CAG TT	AGG GAA GGC CGT GGT TGT
*Il-1β*	GTC ACA AGA AAC CAT GGC ACA T	GCC CAT CAG AGG CAA GGA
*Gapdh*	CAT GGC CTT CCG TGT TCC TA	GCG GCA CGT CAG ATC CA

Pparγ, peroxisome proliferator-activated receptor gamma; Cebpα, CCAAT/enhancer-binding protein alpha; aP2, adipocyte fatty acid-binding protein; Pgc1α, Pparγ coactivator 1 alpha; Il-6, interleukin 6; Il-1β, interleukin 1β; Gapdh, glyceraldehyde 3-phosphate dehydrogenase.

**Table 6 ijms-23-06384-t006:** List of antibodies for Western blot analysis.

	Antibody	Dilution Factor	Corporation	Catalog Number
Primary antibody	p-JNK	1:500	Cell Signaling	9251
	p-ERK	1:3000	Cell Signaling	4370
	BiP	1:1000	Cell Signaling	3183
	CHOP	1:1000	Cell Signaling	2895
	HO-1	1:1000	Cell Signaling	5853S
	β-actin	1:2000	Santa Cruz	sc-47778
Secondary antibody	Anti-rabbit IgG	1:3000	Cell Signaling	7074
	Anti-mouse IgG	1:1000	Cell Signaling	7076

p-JNK, phospho-stress-activated protein kinases/Jun-amino-terminal kinases SAPK/JNK; p-ERK, phospho-extracellular signal-regulated kinases; BiP, binding immunoglobulin protein; CHOP, C/EBP homologous protein; HO-1, heme oxygenase (decycling) 1; IgG, immunoglobulin g.

**Table 7 ijms-23-06384-t007:** Statistical summary.

Parameter	LPS Main Effect	Diet Main Effect	LPS × Diet Interaction
**Relative tissue weights**			
Liver	ns *p* = 0.315	ns *p* = 0.422	ns *p* = 0.616
Epididymal adipose tissue	ns *p* = 0.940	ns *p* = 0.170	ns *p* = 0.783
Mesenteric adipose tissue	ns *p* = 0.497	ns *p* = 0.877	ns *p* = 0.951
Retroperitoneal adipose tissue	ns *p* = 0.214	ns *p* = 0.359	ns *p* = 0.537
Perirenal adipose tissue	*** *p* < 0.001	ns *p* = 0.591	ns *p* = 0.777
White adipose tissue	ns *p* = 0.630	ns *p* = 0.092	ns *p* = 0.555
**Serum lipid profiles**			
Triglyceride	**** *p* < 0.0001	ns *p* = 0.795	ns *p* = 0.732
Total cholesterol	**** *p* < 0.0001	ns *p* = 0.292	* *p* < 0.05
HDL-cholesterol	**** *p* < 0.0001	** *p* < 0.01	* *p* < 0.05
LDL-cholesterol	**** *p* < 0.0001	* *p* < 0.05	** *p* < 0.01
Cardiac risk factor	**** *p* < 0.0001	*** *p* < 0.001	*** *p* < 0.001
**Serum inflammatory mediators**			
IL-1β	**** *p* < 0.0001	**** *p* < 0.0001	**** *p* < 0.0001
IL-10	**** *p* < 0.0001	**** *p* < 0.0001	**** *p* < 0.0001
TNF-α	** *p* < 0.01	*** *p* < 0.001	* *p* < 0.05
CXCL-1	**** *p* < 0.0001	* *p* < 0.05	* *p* < 0.05

* *p* < 0.05, ** *p* < 0.01, *** *p* < 0.001, **** *p* < 0.0001, ns = not significant. LPS, Lipopolysaccharide; HDL, high-density lipoprotein; LDL, low-density lipoprotein; IL-1β, interleukin 1β; IL-10, interleukin 10; TNF-α, tumor necrosis factor alpha; CXCL-1, chemokine (C-X-C motif) ligand 1.

## Data Availability

The data presented in this study are available from the corresponding authors upon request.

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
