# Peer review of "Lowering n-6/n-3 Ratio as an Important Dietary Intervention to Prevent LPS-Inducible Dyslipidemia and Hepatic Abnormalities in ob/ob Mice"

_ijms, 2022, doi:10.3390/ijms23126384_

Round 1

Reviewer 1 Report

My sincere congratulations on a very good manuscript. Assessment of the influence of the partial replacement of SFAs with PO or CO on attenuate metabolic complications in genetically obese mice is very interesting. I am glad that the Authors have taken up this research topic.

Overall merit is very good. However, the non-classical manuscript presentation (introduction, material and methods, results, discussion) makes it somewhat difficult to understand.
The manuscript is quite long, therefore it is important to keep it transparent. Hence it seems that
- short introductions, such as, for example, To investigate the effects of dietary FAs replacement with either PO or CO on blood lipid profiles, total FAs lipid profiles in the blood of experimental mice were analyzed by gas chromatography, should move to the material and methods section
- descriptions such as: Obesity-induced MetS generally exhibits detrimental effects on glucose metabolism due to the dyslipidemia and inflammation [38] and eventually leads to the development of insulin resistance in ob / ob mice [39] OR Previous studies have revealed that dyslipidemia is highly associated with NAFLD in patients who have obesity [40,41]. Higher levels of serum LDL-C are significantly correlated with inflammatory responses and hepatic damages [42.43] should be removed entirely
- so it describes in detail the dietary recommendations (along with a table) in the discussion that are not recommended. It is better to write briefly what the Authors really wanted to convey
- table descriptions (in particular Tables 1, 2 and 3) should be corrected - in which units are the result expressed (g? or%?), which means a? b? C? looking only at the Tables it is unclear

Author Response

We highlighted our updates with green (for reviewer 1) and red (for reviewer 2). Furthermore, all the authors re-read the entire manuscript and updated grammar and formatting issues with yellow.

[C1] My sincere congratulations on a very good manuscript. Assessment of the influence of the partial replacement of SFAs with PO or CO on attenuate metabolic complications in genetically obese mice is very interesting. I am glad that the Authors have taken up this research topic. Overall merit is very good. However, the non-classical manuscript presentation (introduction, material and methods, results, discussion) makes it somewhat difficult to understand.

[A1] All authors appreciate generous comments, and we fully agreed with your concern concerning the way of manuscript presentation. However, we followed the IJMS format sequence of the manuscript; therefore, there is no room for the update for this issue. Please generously understand the situation.

[C2] The manuscript is quite long, therefore it is important to keep it transparent. Hence it seems that - short introductions, such as, for example, To investigate the effects of dietary FAs replacement with either PO or CO on blood lipid profiles, total FAs lipid profiles in the blood of experimental mice were analyzed by gas chromatography, should move to the material and methods section.

[A2; L161] Authors agree with the reviewer’s comment. Therefore, we determined to delete the redundant expression.

[C3] - descriptions such as: Obesity-induced MetS generally exhibits detrimental effects on glucose metabolism due to the dyslipidemia and inflammation [38] and eventually leads to the development of insulin resistance in ob / ob mice [39] OR Previous studies have revealed that dyslipidemia is highly associated with NAFLD in patients who have obesity [40,41]. Higher levels of serum LDL-C are significantly correlated with inflammatory responses and hepatic damages [42.43] should be removed entirely.

[A2; L183] Authors agree with the reviewer’s comment. Therefore, we determined to delete the redundant expression.

[C4] - so it describes in detail the dietary recommendations (along with a table) in the discussion that are not recommended. It is better to write briefly what the Authors really wanted to convey

[A4] We would like to claim the amount of PUFA and the lower ratio of n-6/n-3 PUFA used in the current study are within the range of the diet recommendation in multiple countries. In addition, these summaries would be helpful to the potential readers who may wish to acknowledge the contemporary discussion of fat and PUFA intake.  Therefore, we wish to keep the original.

[C5] - table descriptions (in particular Tables 1, 2 and 3) should be corrected - in which units are the result expressed (g? or%?), which means a? b? C? looking only at the Tables it is unclear.

[A5; L181-182, 195-196, 245, 272, 303, 309, 334, 339, 377, 395] It is a great comment to update our manuscript. The unit in Table 1 is gram (g) and the fatty acids profile in Table 2 and Table 3 is represented as a percentage (%); therefore, the tables are updated accordingly to the reviewer’s comment. Now authors clearly demonstrated the a,b, and c systems for the statistical analyses in all tables and figures.

Reviewer 2 Report

Park et al. in the article "Lowering n-6/n-3 ratio as an important dietary intervention to prevent LPS-inducible dyslipidemia and hepatic abnormalities in ob/ob mice" investigated the effect of the changed ratio of n-6/n-3 PUFAs and increasing the intake of n-3 PUFAs. The authors provided a study with interesting results. 

My main concern relates to the MUFA content in the diet. Does different MUFA content in the CO diet not affect the observed results? 

In the received manuscript there were no line numbers, therefore referring to a specific fragment was difficult. I provided my specific comments below:

- abstract: LPS - no explanation of the abbreviation

- introduction: to describe the 'two-hit' hypothesis model the authors cited a paper that describes this model as obsolete and inadequate. Does it feel like a good choice of literature?

- many abbreviations are not explained until section 4, please revise the manuscript and include explanations of abbreviations as they first appear, 

- introduction paragraph 4: n-3 PUFA enriched

- introduction paragraph 4: "Most PUFAs are essential FAs because they cannot be synthesized in the human body" - most PUFA? Only two fatty acids are known to be essential for humans

- introduction paragraph 5: "decreases blood TG, TC, and LDL-C levels in animals and humans" - "animals" suggests that research was conducted on a variety of species, the cited article focused only on mice

- introduction paragraph 5: "PO is considered a source of n-3 PUFAs" how many, what type?

- introduction paragraph 6: "Previously, we have reported that the partial replacement of HFD with krill oil (KO) or CO could alleviate high fat-induced obesity, dyslipidemia, insulin resistance, and hepatic steatosis" - in animals? in humans?

- section 2.1: "low n-3 ratio in the CO" - n-3/n-6 ratio?

- table 1: please consider rearranging the manuscript to put tables and figures on one page to make them easier to read

- figure 1: no description under the figure as to what each "*" means, no description under the figure as to what "a, b" means, "Means with different letters (a-b) represent significant differences" - between what is a statistically significant difference for a and b. (note to consider for other tables and figures)

- figure 3: what values are shown on the box and whisker plots? The whiskers can stand for several other things, such as The minimum and the maximum value of the data set, One standard deviation above and below the mean of the data set, The 9th percentile and the 91st percentile of the data set, The 2nd percentile and the 98th percentile of the data set. What does the "+" in the middle of the box mean?

- table 3: what are the units?

- figure 6: What do the black and red arrows mean?

- figure 7: "Same letters mean there’s no statistically significant differences." unclear sentence

- discussion, paragraph 2: "Taken together, the consumption of PUFAs is more important than reducing the ratio of n-6/n-3 PUFA to regulate systemic dyslipidemia in our experimental settings" - consumption of n-3 PUFA?

- discussion, paragraph 3: "adipokine secretion" - which adipokine? adipokines in general?

- section 4.4, 4.5, 4.6 - please describe in brief

Author Response

We highlighted our updates with green (for reviewer 1) and red (for reviewer 2). Furthermore, all the authors re-read the entire manuscript and updated grammar and formatting issues with yellow.

Park et al. in the article "Lowering n-6/n-3 ratio as an important dietary intervention to prevent LPS-inducible dyslipidemia and hepatic abnormalities in ob/ob mice" investigated the effect of the changed ratio of n-6/n-3 PUFAs and increasing the intake of n-3 PUFAs. The authors provided a study with interesting results. 

[C1] My main concern relates to the MUFA content in the diet. Does different MUFA content in the CO diet not affect the observed results?

[A1] That is a crucial comment. PO and CO have more MUFA than ND, but there is no difference between PO and CO groups in MUFA contents. Therefore, when it compares to the differences between PO and CO, MUFA effects were excluded. MUFA may have a significant role in metabolically complicated conditions (i.e. obesity, dyslipidemia, and so on) but, we did not collect enough evidence to make a strong conclusion about the effects of MUFA. Therefore, we did not mention any of the impacts of MUFAs. However, in the future, the effects of MUFA should be clarified in depth.

[C2] In the received manuscript there were no line numbers, therefore referring to a specific fragment was difficult. I provided my specific comments below:

[A2] We apologize for any inconvenience and now we added a line number for clear understanding.

[C3] - abstract: LPS - no explanation of the abbreviation.

[A3; L30-31] Thank you. We added the full name for LPS for the coherence of our manuscript per the reviewer’s comment. Also, we added the full name of other words.

[C4] - introduction: to describe the 'two-hit' hypothesis model the authors cited a paper that describes this model as obsolete and inadequate. Does it feel like a good choice of literature?

[A4; L77, 79, 733-735] Authors really appreciated your crucial comments; therefore, we updated references (#13 and #14).

- Day CP, James OF. Steatohepatitis: a tale of two ‘hits’?, Gastroenterology, 1998, vol. 114 (pg. 842-5)

- Day CP. From fat to inflammation. Gastroenterology. 2006 Jan;130(1):207-10.

[C5] - many abbreviations are not explained until section 4, please revise the manuscript and include explanations of abbreviations as they first appear,

[A5; L28-30, 36-37, 44-45, 89, 93, 96, 98-99, 100, 101, 107, 108, 137, 289-290, 341, 346, 355, 363, 364-365] We completely agree with the reviewer’s comment; therefore, we updated our manuscript.

[C6] - introduction paragraph 4: n-3 PUFA enriched

[A6; L84] We corrected n-3 PUFA enriched with n-3 PUFA enriched.

[C7] - introduction paragraph 4: "Most PUFAs are essential FAs because they cannot be synthesized in the human body" - most PUFA? Only two fatty acids are known to be essential for humans.

[A7; L81] Thank you very much. The authors agree entirely with the reviewers’ comments. We deleted ‘Most’.

[C8] - introduction paragraph 5: "decreases blood TG, TC, and LDL-C levels in animals and humans" - "animals" suggests that research was conducted on a variety of species, the cited article focused only on mice.

[A8; L99 Authors appreciate the in-depth review. We updated our manuscript from animals to mice.

[C10] - introduction paragraph 5: "PO is considered a source of n-3 PUFAs" how many, what type?

[A10; L100] We updated our manuscript with the detailed information.

[C11] - introduction paragraph 6: "Previously, we have reported that the partial replacement of HFD with krill oil (KO) or CO could alleviate high fat-induced obesity, dyslipidemia, insulin resistance, and hepatic steatosis" - in animals? in humans?

[A11; L110-112] It was from rats and we updated it.

[C12] - section 2.1: "low n-3 ratio in the CO" - n-3/n-6 ratio?

[A12; L133] We apologize for any inconvenience for this issue. Now we updated for the clear expression.

[C13] - table 1: please consider rearranging the manuscript to put tables and figures on one page to make them easier to read

[A13] We really appreciate your kind comments. If our manuscript is accepted then we improve our visually since further updates may require extra work.

[C14] - figure 1: no description under the figure as to what each "*" means, no description under the figure as to what "a, b" means, "Means with different letters (a-b) represent significant differences" - between what is a statistically significant difference for a and b. (note to consider for other tables and figures)

[A14] We really appreciate your generous comment. Now we updated the whole captions and we hope our updates are acceptable to the reviewer.

[C15] - figure 3: what values are shown on the box and whisker plots? The whiskers can stand for several other things, such as the minimum and the maximum value of the data set, One standard deviation above and below the mean of the data set, The 9th percentile and the 91st percentile of the data set, The 2nd percentile and the 98th percentile of the data set. What does the "+" in the middle of the box mean?

[A15; L669-671] Missing information was our mistake and thank you for your kind comment. Box-and-Whisker plots depicted with the minimum, the lower (25th percentile), the median (50th percentile), the upper (75th percentile), and the maximum ranked sample, and the average of the assigned group is indicated by a “+” sign. This information is added in the 4.10 section (statistical analysis).

[C16] - table 3: what are the units?

[A16] It represents the percentage (%) and I added it to the table.

[C17] - figure 6: What do the black and red arrows mean?

[A17; L302-303] We clarified the arrows in figure caption 6. The black arrow represents lipid droplets and the red arrow represents lobular inflammation. Now, this information is available in the figure legend.

[C18] - figure 7: "Same letters mean there’s no statistically significant differences." unclear sentence

[A18] We deleted the ambiguous expression. Thank you.

[C19] - discussion, paragraph 2: "Taken together, the consumption of PUFAs is more important than reducing the ratio of n-6/n-3 PUFA to regulate systemic dyslipidemia in our experimental settings" - consumption of n-3 PUFA?

[A20; L419-421] All authors appreciate your comment. We updated our discussion and we hope it is acceptable.

[C20] - discussion, paragraph 3: "adipokine secretion" - which adipokine? adipokines in general?

[A20; L423] We added observed adipokine for clear understanding (i.e. TNF-α, IL-6, and IL-10).

[C21] - section 4.4, 4.5, 4.6 - please describe in brief.

[A21; L579-581, 582-583, 588-590, 595-599] Thank you we updated our manuscript per the reviewers’ comment.